# A Cramér Distance perspective on Non-crossing Quantile Regression in Distributional Reinforcement Learning

## Abstract

Distributional reinforcement learning (DRL) extends the value-based approach by using a deep convolutional network to approximate the full distribution over future returns instead of the mean only, providing a richer signal that leads to improved performances. Quantile-based methods like QR-DQN project arbitrary distributions onto a parametric subset of staircase distributions by minimizing the 1-Wasserstein distance, however, due to biases in the gradients, the quantile regression loss is used instead for training, guaranteeing the same minimizer and enjoying unbiased gradients. Recently, monotonicity constraints on the quantiles have been shown to improve the performance of QR-DQN for uncertainty-based exploration strategies. The contribution of this work is in the setting of fixed quantile levels and is twofold. First, we prove that the Cramér distance yields a projection that coincides with the 1-Wasserstein one and that, under monotonicity constraints, the squared Cramér and the quantile regression losses yield collinear gradients, shedding light on the connection between these important elements of DRL. Second, we propose a novel non-crossing neural architecture that allows a good training performance using a novel algorithm to compute the Cramér distance, yielding significant improvements over QR-DQN in a number of games of the standard Atari 2600 benchmark.

## 1 Introduction

Distributional Reinforcement Learning (DRL) extends the value-based approach of DQN [21] by considering the full distribution of returns as a learning signal allowing to take into account all the complexity of the randomness coming from the rewards, the transitions and the policy, which is hidden when considering the mean only. Even when a policy aims at maximizing the expected return, considering the full distribution provides an advantage in the presence of approximations, allowing to learn better representations and helping to reduce state aliasing [1]. With this new approach comes a generalization of the *Bellman operator*—the *distributional* Bellman operator—, whose contraction properties are key for guaranteeing the stability of DRL algorithms.

How distributions are represented and learned is also a key point in DRL, since some choices can break the contraction property (see, e.g., [25, Lemma 2]). Some approaches use staircase parametric representations whose steps correspond to fixed quantile values like in C51 [1] or to fixed quantile levels like in QR-DQN [10]. Alternatively, FQN [30] fully parameterize the staircase distributions. IQN [9] follows a different approach by approximating the quantile function with a neural network that takes the quantile level as input and must therefore be sampled during training.

DRL methods resort to different notions of distance or divergences between distributions in order to practically learn the distributions but also to analyze the effect on the contraction property of the distributional Bellman operator. In [25], a Hilbert space endowed with the $\ell_2$ norm on cumulative distribution functions has been shown to be a natural framework to analyze the effect of the fixed quantile value representation of C51. In [2], the squared $\ell_2$ distance, called *Cramér distance* in that work,[1] has been proposed for Generative Adversarial Networks but also for machine learning in general due its unbiased gradients. However, attempts to use the Cramér distance as loss function in DRL, yielded results that are inferior to those obtained with the heuristic Kullback-Leibler divergence loss used in C51, as reported in [3]. In [10], the Wasserstein distance has been used for defining how a general distribution should be represented with fixed quantile levels and also to analyze the effect on the contraction property of the distributional Bellman operator. However, due to the biased gradients of the Wasserstein distance, the quantile regression loss is used to train the network, guaranteeing the same minimizer as the 1-Wasserstein distance and enjoying unbiased gradients.

When estimating multiple quantiles, one faces the issue of crossing quantiles, i.e., a violation of the monotonicity of the quantile function. In DRL, crossing quantiles make the learning signal noisy, affecting disambiguation of states as shown in [31]. This issue has been addressed in the statistical literature of quantile regression (see, e.g. [17, 14, 19, 13, 11, 7]) but also, more generally, in the machine learning literature on how to represent and learn monotonic functions (see, e.g., [12, Table 1]), with different approaches like including penalties in the loss function or enforcing monotonicity by design. Methods that take sampled quantile levels as input during training like [27] or [9], have been shown to alleviate the problem. In the DRL literature, [31] enforces monotonicity with a special neural network design obtaining improved results with respect to QR-DQN, in the setting of uncertainty-based exploration.

In this work, we analyze the theoretical properties of the Cramér distance for learning staircase distributions with fixed quantile levels and we propose a practical algorithmic solution for DRL with the standard $\varepsilon$-greedy exploration strategy. In Section 2, we expose the necessary background and definitions. In Section 3, we present the connections between the Cramér distance, the 1-Wasserstein distance and the Quantile Regression loss, leading to a contraction guarantee. In Section 4, we propose a low-complexity algorithm to compute the Cramér distance. In Section 5, we introduce a novel neural network architecture enforcing non-crossing quantiles. In Section 6, we report experimental results on the Atari 2600 benchmark using the Cramér distance algorithm and the proposed architecture. Finally, in Section 7, we give some concluding remarks. Due to space constraints, complete proofs are presented in the supplementary material.

## 2   Background

We consider the classical model of agent-environment interactions [23], i.e., a Markov Decision Process (MDP) $(\mathcal{S}, \mathcal{A}, \mathbb{R}, p, \gamma)$, with $\mathcal{S}$ and $\mathcal{A}$ being the state and action space, $R : \mathcal{S} \times \mathcal{A} \to \mathbb{R}$ being the reward function, $P(s'|s, a) : \mathcal{S} \times \mathcal{A} \times \mathcal{S} \to [0, 1]$ being the probability of transitioning from state $s$ to state $s'$ after taking action $a$ and $\gamma \in [0, 1)$ the discount factor. A stochastic policy $\pi(\cdot|x) : \mathcal{S} \times \mathcal{A} \to [0, 1]$ maps a state $s$ to a distribution over $\mathcal{A}$.

### 2.1   Q-Learning

For a fixed policy $\pi$, the *return* $Z^\pi(s, a)$ is a random variable representing cumulative rewards the agent gains from $(s, a)$ by following the policy $\pi$, e.g. $Z^\pi(s, a) \equiv \sum_{t=0}^{\infty} \gamma^t R(s_t, a_t)$ with $s_0 = s, a_0 = a$ and $s_{t+1} \sim p(\cdot \mid s_t, a_t), a_t \sim \pi(\cdot \mid s_t)$. The usual goal in reinforcement learning (RL) is to find an optimal $\pi^*$ maximizing the expectation of $Z^\pi$, i.e. the state-action value function $Q^\pi(x, a) \equiv \mathbb{E} Z^\pi(s, a)$. *Q-Learning* [29] is an off-policy reinforcement learning algorithm that directly learns the optimal action-value function using the *Bellman optimality operator*

$$(\mathcal{T}Q)(x, a) \equiv \mathbb{E}R(x, a) + \gamma \mathbb{E}_P \max_{a' \in \mathcal{A}} Q(x', a') \tag{1}$$

In the evaluation case, the *Bellman operator* $\mathcal{T}^\pi$ [5, 29] is defined as

$$(\mathcal{T}^\pi Q)(x, a) \equiv \mathbb{E}R(x, a) + \gamma \mathop{\mathbb{E}}_{P, \pi} Q(x', a'). \tag{2}$$

---

[1] In this work, we follow [25] and use the term Cramér distance for the $\ell_2$ distance.

They are contraction mappings and their repeated application to some initial $Q_0$ converges exponentially to $Q^\pi$ or $Q^*$, respectively [6]. However, when $Q$ is represented by a neural network that is trained on batches of sampled transitions $(s, a, r, s')$ as in most deep learning studies, a gradient update is preferred since it allows for the dissipation of noise introduced in the target by stochastic approximation [6, 18]. DQN [21] iteratively trains the network by minimizing the squared *temporal difference (TD)* error $\frac{1}{2}\left[r + \gamma \max_{a'} Q_{\omega^-}(s', a') - Q_\omega(s, a)\right]^2$ over samples $(s, a, r, s')$, where $\omega^-$ is the target network, which is a copy of $\omega$, synchronized with it periodically. When using an $\varepsilon$-*greedy policy*, the samples are obtained while the agent interacts with the environment choosing actions uniformly at random with probability $\varepsilon$ and otherwise according to $\arg\max_a Q_\omega(s, a)$.

## 2.2 Distributional reinforcement learning

In order to extend the previous concepts to distributional reinforcement learning, the distributional Bellman operator and *optimality operator* [1] are defined as

$$(\mathcal{T}^\pi Z)(x, a) \overset{D}{\equiv} R(x, a) + \gamma Z(x', a'), \tag{3}$$

$$(\mathcal{T}Z)(s, a) \overset{D}{\equiv} R(s, a) + \gamma Z\left(s', \arg\max_{a' \in \mathcal{A}} \mathbb{E}_p Z(s', a')\right), s' \sim p(\cdot \mid s, a), a' \sim \pi(\cdot \mid x'), \tag{4}$$

where $Y \overset{D}{\equiv} U$ denotes equality of probability laws, that is the random variable $Y$ is distributed according to the same law as $U$. In order to characterize the contraction properties of these operators, some notion of distance between indexed collections of distributions is necessary. The *p-Wasserstein distance* between distributions $U$ and $Y$ is defined as the $\ell_p$ metric on inverse cumulative distribution functions (inverse CDFs) [22] i.e.

$$d_p(U, Y) = \left(\int_0^1 \left|F_Y^{-1}(\omega) - F_U^{-1}(\omega)\right|^p d\omega\right)^{1/p}$$

where for a random variable $Y$, the *inverse CDF* $F_Y^{-1}$ of $Y$ is defined by

$$F_Y^{-1}(\omega) := \inf\{y \in \mathbb{R} : \omega \leq F_Y(y)\}$$

where $F_Y(y) = \Pr(Y \leq y)$ is the CDF of $Y$.[2] Then the maximal Wasserstein metric between two indexed collections of distributions $Z_1$ and $Z_2$ is defined as $\bar{d}_p(Z_1, Z_2) := \sup_{x,a} d_p(Z_1(x, a), Z_2(x, a))$. [1, Lemma 3] shows that $\mathcal{T}^\pi$ is a contraction in $\bar{d}_p$, i.e.,

$$\bar{d}_p(\mathcal{T}^\pi Z_1, \mathcal{T}^\pi Z_2) \leq \gamma \bar{d}_p(Z_1, Z_2). \tag{5}$$

The case of the distributional optimality operator $\mathcal{T}$ is more involved. In general, it is not a contraction [1]. However, based on the fact that $\mathcal{T}^\pi$ is a contraction, [1] proves that, if the optimal policy is unique, then the iterates $Z_{k+1} \leftarrow \mathcal{T}Z_k$ converge to $Z^{\pi^*}$ (in $p$-Wasserstein metric, $\forall s, a$) and, under some conditions, $\mathcal{T}$ has a unique fixed point corresponding to an optimal value distribution.

## 2.3 Projecting distributions onto a finite support

Previous approaches of distributional reinforcement learning project return distributions $Z(x, a)$ onto a space of distributions of finite support, by modeling it with a mixture of Diracs over $N$ support points $\theta_i(x, a), i = 1..N$

$$Z_\theta(x, a) := \sum_{i=1}^N p_i(x, a)\delta_{\theta_i(x,a)} \tag{6}$$

which yields a staircase CDF $F_{x,a}(z) \equiv \sum_{i=1}^N p_i(x, a)\mathbb{1}_{z \geq \theta_i(x,a)}$. Different approaches have been followed to parameterize these distributions depending on whether $p_i$ and $\theta_i$ are learned or fixed. In this work, we consider $p_i$ fixed and $\theta_i$ a learned parameter.

In order to analyze how arbitrary distributions are mapped into these finite representations, different projection operators are defined as minimizers of some distance between distribution. For instance,

---

[2]For $p = \infty$, $d_\infty(Y, U) = \sup_{\omega \in [0,1]} \left|F_Y^{-1}(\omega) - F_U^{-1}(\omega)\right|$.

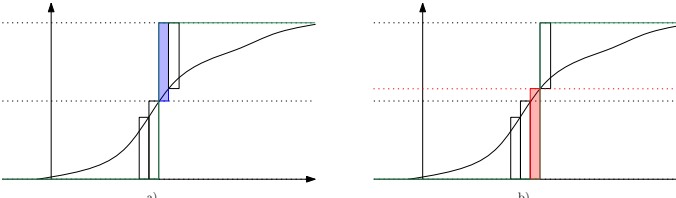

Figure 1: **Midpoint minimizer.** a) The curve is approximated by one Dirac (green curve) located at inverse of the mid-point. The rectangles represent an approximation of $\ell_p$ distance. b) If we move the Dirac in one direction or the other, the blue rectangle will be replaced by a larger one (in red here).

in [10], the 1-Wasserstein projection $\Pi_{W_1}$ is used and it is shown that the resulting projected Bellman operator remains a contraction, i.e.,

$$\bar{d}_\infty\left(\Pi_{W_1}\mathcal{T}^\pi Z_1, \Pi_{W_1}\mathcal{T}^\pi Z_2\right) \leq \gamma \bar{d}_\infty\left(Z_1, Z_2\right). \tag{7}$$

However, since Wasserstein distances suffer from biased gradients [2, 1], the *quantile regression (QR) loss* is used in practice, guaranteeing the same minimizer and enjoying unbiased gradients [10]. The QR loss for learning the parameters $\{\theta_1, \ldots, \theta_N\}$ of $F(z) \equiv \frac{1}{N}\sum_{i=1}^N \mathbb{1}_{z \geq \theta_i}$ given a target $\bar{F}$ is

$$\mathcal{L}_{\mathrm{QR}}(F, \bar{F}) \equiv \sum_{i=1}^N \frac{1}{N} \mathbb{E}_{Z \sim \bar{F}}\left[\rho_{\hat{\tau}_i}(Z - \theta_i)\right] \text{ with } \rho_\theta(u) \equiv u(\tau - \mathbb{1}_{u<0}) \tag{8}$$

where $\hat{\tau}_i$ are the midpoints of a uniform grid of $N$ quantile levels, i.e. $\hat{\tau}_i \equiv \frac{2i-1}{2N}$. Note that this definition makes $\theta_i$ an estimate of the $\hat{\tau}_i$-quantile. As we shall see, this correspondence will not be enforced by the Cramér projection we consider next.

## 3 Theoretical results for the Cramér projection

Motivated by the practical interest of the unbiased gradients of the squared Cramér distance[3] [2] $\int_{-\infty}^\infty (F(z) - \bar{F}(z))^2 dz$, we now analyse theoretical properties related to minimizing this quantity in order to define the Cramér projection and highlight the connections with the 1-Wassertein projection and the QR loss.

### 3.1 Equivalence with the 1-Wasserstein projection

We now show that, given an arbitrary distribution and a grid of quantile levels, there is a staircase representation that minimizes the $\ell_p$ distance, which puts the quantile values at the inverse of the quantile level midpoints.

**Theorem 1.** *Given $p_i \geq 0, i = 1..N$ such that $\sum_i p_i = 1$, the $\ell_p$ distance between $F$ and a mixture of Heaviside step functions $F_N(z) = \sum_{i=1}^N p_i \mathbb{1}_{z \geq \theta_i}$ is minimized with $\theta_i = F^{-1}((\tau_i + \tau_{i-1})/2)$ where $\tau_i$ are the quantile levels $\tau_i = \sum_{j=1}^i p_j$.*

*Proof sketch.* The proof is based on the observation illustrated in Fig. 1. See Appendix for details. $\square$

**Remark 1.** *For simplicity, we chose $\theta_i = F^{-1}((\tau_i + \tau_{i-1})/2)$, however any permutation $\sigma$ in the symmetric group of size $N$ makes $\tilde{\theta}_i \equiv \theta_{\sigma(i)}$ a minimizer too.*

We define the $\ell_p$ projection of an arbitrary CDF $F$ with inverse $F^{-1}$ onto a grid of quantile levels as

$$\Pi_{\ell_p} F \equiv F_N^\star(z) = \sum_{i=1}^N p_i \mathbb{1}_{z \geq \theta_i^\star} \text{ with } \theta_i^\star = F^{-1}((\tau_i + \tau_{i-1})/2). \tag{9}$$

Therefore, it is equivalent to the 1-Wasserstein projection (see [10, Lemma 2]). This directly implies that the Cramér projected Bellman operator is also a contraction.

---

[3]As in [25], we call Cramér distance the $\ell_2$ distance and Cramér loss its square.

**Corollary 1.** *The Cramér projected distributional Bellman operator is a contraction in $\bar{d}_\infty$ i.e.*

$$\bar{d}_\infty \left( \Pi_{\ell_p} \mathcal{T}^\pi Z_1, \Pi_{\ell_p} \mathcal{T}^\pi Z_2 \right) \leq \gamma \bar{d}_\infty \left( Z_1, Z_2 \right). \tag{10}$$

*Proof.* It follows directly from (7) [1, Lemma 3] and Lemma 1. □

## 3.2 Collinearity of QR loss and Cramér gradients under non-crossing constraints

In order to put in evidence the relationship betweem the gradients of the QR and Cramér loss, we first present an alternative formula for the Cramér loss.

**Lemma 1.** *Given two staircase distributions $F(z) = \frac{1}{N} \sum_{i=1}^N \mathbb{1}_{z \geq \theta_i}$ and $\bar{F}(z) = \frac{1}{N} \sum_{i=1}^N \mathbb{1}_{z \geq \bar{\theta}_i}$ such that $\theta_1 < \cdots < \theta_N$ and $\bar{\theta}_1 < \cdots < \bar{\theta}_N$. Let $u_{ij} \equiv \bar{\theta}_j - \theta_i$ and $\delta_{ij} \equiv \mathbb{1}_{u_{ij} < 0}$. The squared Cramér distance between the distributions can be expressed as*

$$\int_{-\infty}^{\infty} (F(z) - \bar{F}(z))^2 dz = \frac{1}{N^2} \sum_{i=1}^N |u_{ii}| + 2 \left( \sum_{j=i+1}^N \delta_{ij} |u_{ij}| + \sum_{j=1}^{i-1} (1 - \delta_{ij}) |u_{ij}| \right). \tag{11}$$

*Proof sketch.* We resort to a tiling operator to break the integral into pieces. Our demonstration unfold through these steps; First, we prove formally that our operator is well built: the sum of the tiling measured with the operator $\rho$ is equal to the Cramér distance between the two curves. Secondly, we derive Eq. (11) by using that tiling operator. See Appendix for full details. □

**Corollary 2.** *For $F(z) \equiv \frac{1}{N} \sum_{i=1}^N \mathbb{1}_{z \geq \theta_i}$ and $\bar{F}(z) \equiv \frac{1}{N} \sum_{i=1}^N \mathbb{1}_{z \geq \bar{\theta}_i}$ we have*

$$\frac{\partial \mathcal{L}_{QR}(F, \bar{F})}{\partial \theta_i} = \frac{1}{N} \left( \frac{1 - 2i}{2} + \sum_{j=1}^N \delta_{ij} \right) \quad and \quad \frac{\partial \ell_2^2(F, \bar{F})}{\partial \theta_i} = \frac{1}{N^2} \left( 1 - 2i + 2 \sum_{j=1}^N \delta_{ij} \right) \tag{12}$$

*where $\delta_{ij} \equiv \mathbb{1}_{u_{ij} < 0}$. Therefore, their gradients are collinear, i.e.*

$$\nabla_\theta \mathcal{L}_{QR} = \frac{N}{2} \nabla_\theta \ell_2^2. \tag{13}$$

*Proof sketch.* The results are obtained by differentiating (8) and (11). See Appendix. □

**Remark 2.** *Therefore, gradient descent methods whose parameter updates are invariant to rescaling of the gradient like ADAM [16], yield the same optimization path with both losses.*

**Remark 3.** *Huberization of the QR loss (see [10]) breaks the equivalence with the Cramér loss.*

# 4 A low-complexity algorithm for computing the Cramér distance

The formula (11) allows to compute the squared Cramér distance—which we refer to as *Cramér loss*—between two staircase distributions $F(z) = \frac{1}{N} \sum_{i=1}^N \mathbb{1}_{z \geq \theta_i}$, and $\bar{F}(z) = \frac{1}{N} \sum_{i=1}^N \mathbb{1}_{z \geq \bar{\theta}_i}$ assuming the quantiles are ordered, i.e., $\theta_1 < \cdots < \theta_N$ and $\bar{\theta}_1 < \cdots < \bar{\theta}_N$. That formula involves two nested sums making it of quadratic complexity in $N$ as the quantile regression loss. Alternatively, if we consider the sorted sequence of merged quantiles $\theta' \equiv \text{sort} \left( \{\theta_i\}_{i=1..N} \bigcup \{\bar{\theta}_i\}_{i=1..N} \right)$, we have that $F(z) - \bar{F}(z)$ is constant between any two consecutive quantile values of $\theta'$ and the difference can be obtained by accumulating the increments from $F$ and the decrements from $\bar{F}$, see Fig. 2 for an illustration.[4] Therefore, we can express the Cramér loss between two staircase distributions as follows

$$\int_{-\infty}^{\infty} (F(z) - \bar{F}(z))^2 dz = \sum_{i=1}^{2N-1} (\theta'_{i+1} - \theta'_i) \left( \sum_{j \text{ s.t. } \theta_j \leq \theta'_i} \frac{1}{N} - \sum_{j \text{ s.t. } \bar{\theta}_j \leq \theta'_i} \frac{1}{N} \right)^2. \tag{14}$$

In Fig. 2, we propose an algorithm that implements this formula based on sorting the merged quantiles of both distributions, yielding an $O(N \log N)$ complexity. Note that this algorithm does not require the input vectors $\theta$ and $\bar{\theta}$ to be ordered.

---

[4]See Appendix for details.

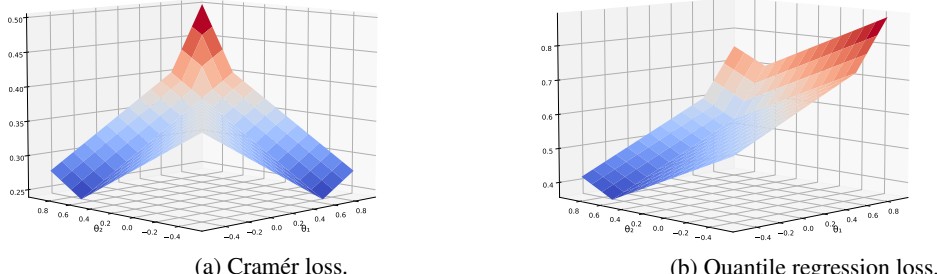

**Algorithm 1: Cramér loss.**

**Input:** $\boldsymbol{\theta} \equiv [\theta_1, \ldots, \theta_N]$,
$\quad\quad \bar{\boldsymbol{\theta}} \equiv [\bar{\theta}_1, \ldots, \bar{\theta}_N]$: array
**Output:** $\int_{-\infty}^{\infty} (F(z) - \bar{F}(z))^2 dz$

$\quad \boldsymbol{\theta}' \leftarrow \text{concat}(\boldsymbol{\theta}, \bar{\boldsymbol{\theta}})$
$\quad i_1, \ldots, i_N \leftarrow \text{argsort}(\boldsymbol{\theta}')$
$\quad \boldsymbol{\theta}' \leftarrow \boldsymbol{\theta}'[i_1, \ldots, i_N]$
$\quad \Delta_z \leftarrow \boldsymbol{\theta}'[1:] - \boldsymbol{\theta}'[:\text{-}1]$
$\quad \Delta_\tau \leftarrow \text{concat}\left(-\frac{1}{N}\mathbf{1}_N, \frac{1}{N}\mathbf{1}_N\right)$
$\quad \Delta_\tau \leftarrow \Delta_\tau[i_1, \ldots, i_N]$
$\quad \Delta_\tau \leftarrow \text{cumsum}(\Delta_\tau)[:\text{-}1]$
$\quad I \leftarrow \Delta_\tau * \Delta_\tau * \Delta_z$
$\quad \textbf{return } \text{sum}(I)$

Figure 2: **Cramér loss algorithm.** The operators $[1:]$ and $[:\text{-}1]$ remove, respectively, the first and the last elements of the array. $\mathbf{1}_N$ denotes an array of $N$ ones and $*$ denotes elementwise multiplication.

(a) Cramér loss.

(b) Quantile regression loss.

Figure 3: **Symmetry in the Cramér loss (left) in comparison to QR loss (right).** The loss landscape correspond to estimating the return distribution of a state $s_0$ with transitions to states $s_1$ and $s_2$ with probability $1/3$ and $2/3$, respectively, whose return distributions are Diracs located at $-0.5$ and $0.6$ respectively, with $N = 3$. The plots are for a fixed $\theta_0 = -0.5$. Notice that when $\theta_0 < \theta_1 < \theta_2$, the two losses have collinear gradients as shown in Corollary 2.

## 5 A centered non-crossing architecture enforcing ordered quantiles

### 5.1 Motivation

When using Algorithm 1, $\theta_i$ is just a location where a mass of $1/N$ is assigned in the estimated distribution, not necessarily corresponding to the $\hat{\tau}_i$-quantile. Therefore, Cramér distance makes the problem of crossing quantiles as usually described vanish, since equivalent distributions can be obtained with $\tilde{\theta}_i \equiv \theta_{\sigma(i)}$, with any permutation $\sigma$ in symmetric group of size $N$. Since the order of quantiles is not constrained using this algorithm, we remove an important constraint on the values that the neural network parameters can take: the domain of parameters that lead to a valid distribution is not reduced. We could therefore expect to be able to reach better results than when using QR loss (QR-DQN) with a same number of parameters. However another problem arises, this permutation equivalence creates symmetries in the loss landscape as illustrated in Fig. 3. These symmetries can hinder the learning process if jumps between symmetric regions occur. In Fig. 5, we show examples of the performance of the QR-DQN network trained with the Cramér loss, denoted as CR-DQN. In particular, in the games of `Breakout` and `Seaquest`, CR-DQN shows a slow learning curve.

For this reason, neural architectures enforcing non-crossing quantiles are also of interest when using the Cramér loss since they drastically reduce the search space by removing the symmetries. The non-crossing architecture of NC-QR-DQN [31] is composed of two subnetworks: one network to estimate extreme quantile for $\tau = 0$ and the scale of the distribution, which is essentially equivalent to estimating the quantile $\tau = 1$, and another network to estimate a grid of quantiles between these extremes using a softmax. As shown by the classical theory of extreme values, extreme quantiles

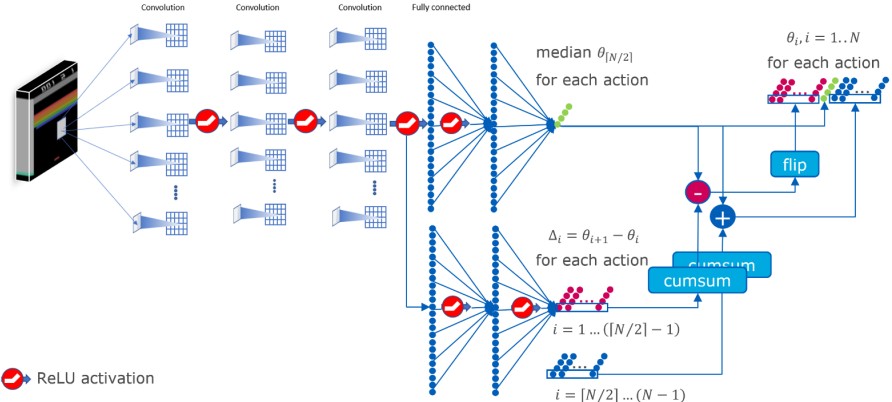

Figure 4: **Centered Non-Crossing (CNC) architecture.**

require more samples to be properly estimated. We hypothesize that this can strongly affect the first stages of the training process where a high error in the estimates for these extreme quantiles propagates to the more centered and easier to estimate quantiles. In the training curves of Fig. 5, we observe indeed a slower training, and no training at all in the case of the game `Seaquest`.

## 5.2 Network description

The previous reasons motivate our novel architecture consisting of two parts: one is dedicated to the median whose estimation is easier and robust and the other estimates the rest of the quantiles by accumulating increments/decrements from the median instead of accumulating from the extreme quantile $\tau = 0$ like in NC-QR-DQN.

Now we formally describe our architecture illustrated in Fig. 4. The first part consists, as in QR-DQN and NC-QR-DQN, of a multi-layer convolutional operator $\mathcal{C}$ (a series of convolutional layers each one followed by a ReLU activation) that is applied to the input state $s$ to obtain the embedded state $e \equiv \mathcal{C}(s) \in \mathbb{R}^d$ given as input to the two subnetworks $\mathcal{M}$ and $\mathcal{D}$ defined next. Let $\mathcal{F}_{\lambda,\eta,D}$ denote a multi-layer fully connected operator with $\lambda$ hidden layers with $\eta$ nodes each one followed by a ReLU activation and a final linear layer of dimension $D$, then the vector of quantiles $\theta(e)$ is obtained as follows

$$\mathcal{M}(e) \equiv \mathcal{F}_{\lambda,\eta,1\times|\mathcal{A}|}(e) \in \mathbb{R}^{1\times|\mathcal{A}|}$$

$$\Delta(e) \equiv \text{ReLU}(\mathcal{F}_{\lambda,\eta,(N-1)\times|\mathcal{A}|}(e)) \in \mathbb{R}^{(N-1)\times|\mathcal{A}|}$$

$$\Delta^-(e) \equiv \Delta(e)[1..(\lceil N/2 \rceil - 1)] \in \mathbb{R}^{(\lceil N/2 \rceil - 1)\times|\mathcal{A}|}$$

$$\Delta^+(e) \equiv \Delta(e)[(\lceil N/2 \rceil..N] \in \mathbb{R}^{(N-\lceil N/2 \rceil)\times|\mathcal{A}|}$$

$$\theta(e) \equiv \text{concat}(\text{flip}(\mathcal{M}(e) - \text{cumsum}(\Delta^-(e))), \mathcal{M}(e), \mathcal{M}(e) + \text{cumsum}(\Delta^+(e))) \in \mathbb{R}^{N\times|\mathcal{A}|}$$

where the operators concat, flip and cumsum operate along the dimension 1 (i.e. corresponding to the quantile index) as well as the operator $[\cdot]$ that extracts the values for the given indices. As in QR-DQN, the mean used to select the best action in an $\varepsilon$-greedy policy is computed as $\frac{1}{N}\sum_{i=1}^{N}\theta_i$.

Fig. 5 shows the improvement achieved by our network on three Atari games.

## 6 Experiments

`DQN_ZOO` **and Atari 2600 benchmark.** We implemented our algorithm on top of the `DQN_ZOO` [24] framework, which integrates reference implementations of RL algorithms with the gym/atari-py RL environment [8]. `DQN_ZOO` provides pre-computed simulation results for each of these algorithms, each of them being run on 5 seeds and on the full set of 57 Atari 2600 games. It was necessary to copy, within atari-py package, a few settings from the Atari Learning Environment [4], in order to make atari-py able to handle the game `Surround`, and to replace a ROM for `Defender`.

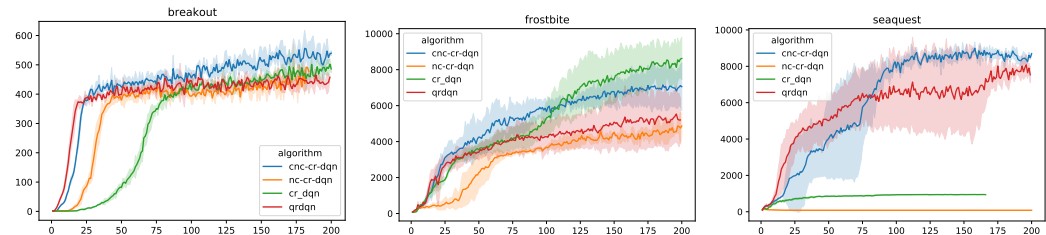

Figure 5: **Comparison to NC-QR-DQN and QR-DQN architectures.** QR-DQN and CR-DQN have exactly the same number of parameters. We denote the NC-QR-DQN architecture with Cramér loss and the standard epsilon-greedy exploration as NC-CR-DQN to avoid confusion with the results reported in [31] that corresponds to the uncertainty-based exploration strategy. Note that given the results of Section 3, it is equivalent to using the QR loss. NC-CR-DQN and CNC-CR-DQN have a number of parameters within $0.1\%$. ADAM optimizer was used with learning rate $5 \times 10^{-5}$ for all.

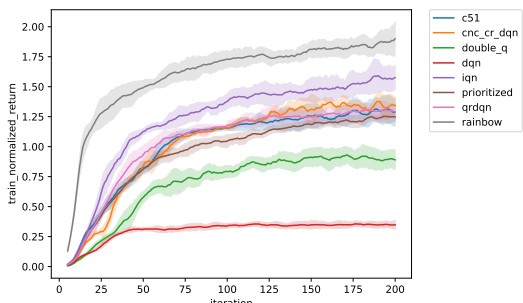

Figure 6: Online training performance, in terms of median human-normalized scores

**Software-hardware setup.**    g4dn.2xlarge AWS machines were used to run some of the experiments. By carefully selecting the pairs of games, it is possible to run two simultaneous experiments, taking around 22 minutes on those machines for two simultaneous iterations. Therefore, for two seeds, 57 games take around 4180 hours of computation. When possible to fit in RAM, three simultaneous runs can be performed, taking around 24 minutes per simultaneous iterations. These times were achieved by restricting each process to one cpu only (processor affinity), otherwise we experienced cpu-gpu communication inefficiencies. These inefficiencies were also experienced when running more simultaneous processes on larger machines. Code and full output of the experiments is available at https://github.com/NB5234123/cnc-cr-dqn.

**Hyperparameters for CNC-CR-DQN**    For model training, we set our hyperparameters with the values used in [10] (epsilon decay, ADAM's $\epsilon$ parameter, experience replay settings). Each experiment consists in 200 iterations. Each iteration is made of a learning phase (1 million frames), followed by an evaluation phase, on 500 thousands frames. We thus use the same experiment procedure, and the same epsilon hyperparameter than the one used for the experiments provided with `DQN_ZOO`; also, our neural network architecture uses the same convolution layers than the other algorithms implemented within `DQN_ZOO`. The experiment settings being the same, our experiment performance can therefore be compared to the experiment data provided with `DQN_ZOO` for the other algorithms. Finally, each of the two heads of our neural network is made of $\lambda = 1$ layer of $\eta = 512$ neurons, with $N = 201$. We obtained the best results with the learning rate set to $5 \times 10^{-5}$. See Appendix for more details.

**Contenders**    Our algorithm is compared to other pure DRL algorithms: C51 [1], QR-DQN [10] and IQN [9]. QR-DQN corresponds to QR-DQN-1, as named in [10], i.e., it is trained via the Huber quantile loss: a quantile loss with a non-linear smoothing approximation when close to 0 parameterized by $\kappa = 1$. We also consider for reference DQN [21], Double Q [28], Prioritized Experience Replay [26] and Rainbow [15]. The latter bases itself on C51 and add many orthogonal improvements like prioritized experience replay. While it is interesting to show their scores beside the ones of pure DRL algorithms, they cannot be used to assess the performance of our algorithm.

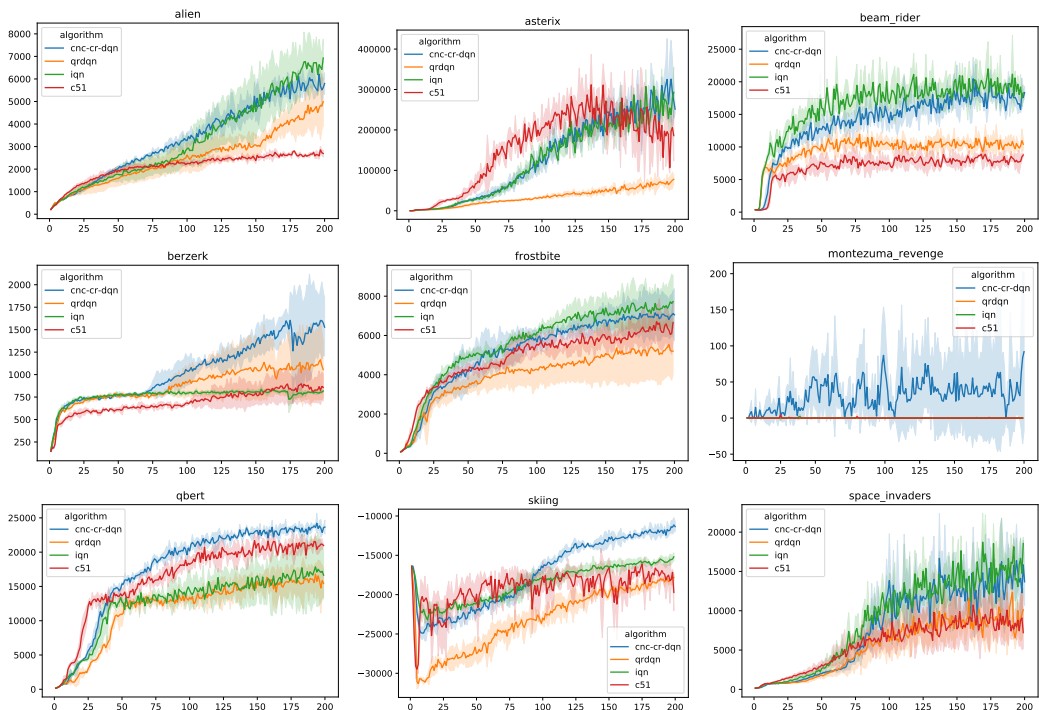

Figure 7: **Training performance.** Curves show average score and standard deviation over three seeds.

**Online training performance** Performance during training protocol: this protocol, described in [20], puts the emphasis on the learning quality. It consists in using normalized training scores to evaluate the algorithms. Fig. 6 shows the median over the 57 games of the human-normalized training scores for each algorithm. The performance curve for an algorithm is obtained as follows: we first measure, for each run of the algorithm and each iteration, the median of human-normalized score; we then calculate the mean score value over the runs, apply a moving average of 5 iterations in order to smooth the curve. On Fig. 6, the shadow areas denote the standard deviation of the scores over the runs. It should be noted that DQN_ZOO provides data for 5 different seeds, for all algorithms and games; while we only did runs with two different seeds using the CRC-CR-DQN algorithm, for each of the games. Human-normalization of score is given by [28]: $\text{normalized\_score} = \frac{\text{agent\_score}-\text{random}}{\text{human}-\text{random}}$ where $\text{random}$ and $\text{human}$ are baseline scores, given for each game. Fig. 7 show some remarkable cases, although these must be taken with caution since the number of seeds is not high enough to draw statistical conclusions. See Appendix for detailed results on the full set of 57 games.

## 7 Conclusion

In this work, we focused on learning staircase distributions on a uniform grid of quantile levels. We showed that learning distributions with the quantile regression loss under non-crossing constraints is essentially equivalent to learning with the Cramér loss. In prior work, crossing quantiles have been studied as a problem for interpretability and state disambiguation. However, when using the Cramér loss on a uniform grid of quantile levels, the problem of crossing quantiles does not exist anymore since quantiles are not identified, representing locations where $1/N$ of the mass is assigned. However, this lack of order generates a permutation equivalence generating symmetries in the loss landscape that hinder the learning process making important the use of non-crossing architectures. On the practical side, we proposed a combination of the Cramér loss and a centered neural architecture enforcing ordered quantiles, yielding significant improvements over QR-DQN in a number of games, sometimes beating all the state-of-the-art pure DRL methods.

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
