# A Appendix

## A.1 Additional theoretical results and proofs

We first prove an auxiliary Lemma.

**Lemma 2.** *For any $\tau, \tau' \in [0,1]$ with $\tau < \tau'$ and cumulative distribution function $F$ with inverse $F^{-1}$, let $t \equiv F^{-1}(\tau)$ and $t' \equiv F^{-1}(\tau')$ and consider the scaled and vertically shifted Heaviside step function $H_\theta^{\tau,\tau'}(z) \equiv \tau + (\tau' - \tau)\mathbb{1}_{z \geq \theta}$. Then, for any $p \in \mathbb{R}, p > 1$, the set of $\theta \in [t, t']$ minimizing*

$$\int_t^{t'} |F(z) - H_\theta^{\tau,\tau'}|^p dz \tag{15}$$

*is given by*

$$\left\{ \theta \in [t, t'] | F(\theta) = \left( \frac{\tau + \tau'}{2} \right) \right\}. \tag{16}$$

*In particular, if $F^{-1}$ is the inverse CDF, then $F^{-1}((\tau + \tau')/2)$ is always a valid minimizer, and if $F^{-1}$ is continuous at $(\tau + \tau')/2$, then $F^{-1}((\tau + \tau')/2)$ is the unique minimizer.*

*Proof.* We decompose the integral as follows

$$\int_t^{t'} |F(z) - H_\theta^{\tau,\tau'}(z)|^p dz = \int_t^\theta (F(z) - \tau)^p dz + \int_\theta^{t'} (\tau' - F(z))^p dz \tag{17}$$

$$= \lim_{a \to t} \int (F(z) - \tau)^p dz \big|_a^\theta + \lim_{b \to t'} \int (\tau' - F(z))^p dz \big|_\theta^b \tag{18}$$

where the limits are taken to cover the particular cases of $t = -\infty$ and $t' = \infty$. Since we are minimizing with respect to $\theta$ we can drop the constant terms and consider

$$\frac{d}{d\theta} \int (F(z) - \tau)^p dz \big|_\theta - \int (\tau' - F(z))^p dz \big|_\theta = (F(\theta) - \tau)^p - (\tau' - F(\theta))^p. \tag{19}$$

First note that for $\theta \in [t, t']$, we have $F(\theta) - \tau > 0$ and $\tau' - F(\theta) > 0$. Then, equating the derivative to zero yields

$$(F(\theta) - \tau)^p - (\tau' - F(\theta))^p = 0 \tag{20}$$

$$\Leftrightarrow F(\theta) - \tau = \tau' - F(\theta) \tag{21}$$

$$\Leftrightarrow F(\theta) = \frac{\tau + \tau'}{2}. \tag{22}$$

By replacing $=$ by $<$ in the previous equations, we see that the sign of the derivative is negative for $\theta < F^{-1}(\frac{\tau+\tau'}{2})$ (since $F$ is increasing) and positive otherwise, which proves the claim. $\square$

**Theorem 1.** *Given $p_i \geq 0, i = 1..N$ such that $\sum_i p_i = 1$, the $\ell_p$ distance between $F$ and a mixture of Heaviside step functions $F_N(z) = \sum_{i=1}^N p_i \mathbb{1}_{z \geq \theta_i}$ is minimized with $\theta_i = F^{-1}((\tau_i + \tau_{i-1})/2)$ where $\tau_i$ are the quantile levels $\tau_i = \sum_{j=1}^i p_j$.*

*Proof.* Let $t_i \equiv F^{-1}(\tau_i)$. We first prove that an optimal $\theta^\star$ satisfies $t_{i-1} \leq \theta_i^\star \leq t_i$. See Fig. 8 for an intuition.

Without loss of generality, we assume that $\theta_1^\star \leq \ldots \leq \theta_N^\star$. Let us suppose that there is an optimal $F_N$ with $\theta_1 \geq t_1$. We can write the $p$-th power of the $\ell_p$ distance as

$$\ell_p^p(F, F_N) = \int_{-\infty}^{t_1} |F(z) - F_N(z)|^p dz + \int_{t_1}^{\theta_2} |F(z) - F_N(z)|^p dz + \int_{\theta_2}^\infty |F(z) - F_N(z)|^p dz \tag{23}$$

The value of the middle term strictly decreases when $\theta_1$ decreases toward $t_1$ (while the other terms are unaffected) since

$$\int_{t_1}^{\theta_2} |F(z) - F_N(z)|^p dz = \int_{t_1}^{\theta_2} |F(z) - H_{\theta_1}^{0,\tau_1}(z)|^p dz \tag{24}$$

$$= \int_{t_1}^{\theta_1} F(z)^p dz + \int_{\theta_1}^{\theta_2} (F(z) - \tau_1)^p dz \tag{25}$$

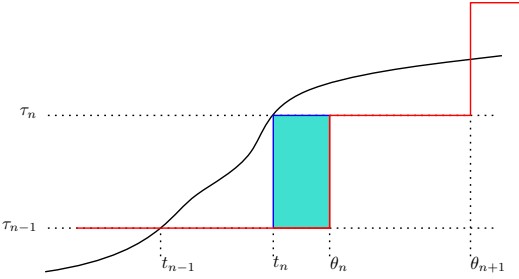

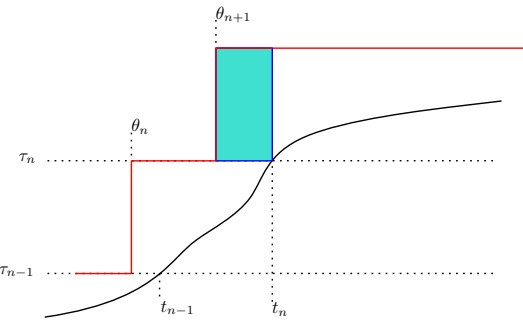

Figure 8: **Intuition for proving** $t_{i-1} \leq \theta_i^\star \leq t_i$. The $\ell_p$ distance can be decreased by moving $\theta_n$ in the first situation and $\theta_{n+1}$ to $t_n$ in the second one. The shaded area represents the improvement for $p = 1$.

and $F(z)^p > (F(z) - \tau_1)^p$. In consequence $\theta_1 = t_1$ ; It proves that no optimal exist for $\theta_1 > t_1$, and thus that we have $\theta_1 \leq t_1$.

By induction, we assume that $\theta_{n-1}^\star \leq t_{n-1}$. As before, we suppose, that there is an optimal $F_N$ with $\theta_n \geq t_n$ and we observe that the value of the term

$$\int_{t_n}^{\theta_{n+1}} |F(z) - F_N(z)|^p dz = \int_{t_n}^{\theta_{n+1}} |F(z) - H_{\theta_n}^{\tau_{n-1},\tau_n}(z)|^p dz \tag{26}$$

$$= \int_{t_n}^{\theta_n} (F(z) - \tau_{n-1})^p dz + \int_{\theta_n}^{\theta_{n+1}} (F(z) - \tau_n)^p dz \tag{27}$$

strictly decreases when $\theta_n$ decreases toward $t_n$ since $(F(z) - \tau_{n-1})^p > (F(z) - \tau_n)^p$. In consequence $\theta_n = t_n$ ; It proves that no optimal exist for $\theta_n > t_n$, and thus that we have $\theta_n \leq t_n \forall n \in \{1..N\}$.

(starting by $\theta_N$ and going backwards). This allows us to show that the optimization problem has an optimal substructure and thus it amounts to solving independent minimization problems of the form (15) i.e.

$$\min_{\theta_1,\ldots,\theta_N} \ell_p^p(F, F_N) = \min_{\theta_1,\ldots,\theta_N} \sum_{i=1}^{N} \int_{t_{i-1}}^{t_i} |F(z) - F_N(z)|^p dz \tag{28}$$

$$= \sum_{i=1}^{N} \min_{\theta_i} \int_{t_{i-1}}^{t_i} |F(z) - H_{\theta_i}^{\tau_{i-1},\tau_i}(z)|^p dz \tag{29}$$

with $t_0 \equiv -\infty$. $\qquad \square$

**Lemma 1.** *Given two staircase distributions $F(z) = \frac{1}{N}\sum_{i=1}^{N} \mathbb{1}_{z \geq \theta_i}$ and $\bar{F}(z) = \frac{1}{N}\sum_{i=1}^{N} \mathbb{1}_{z \geq \bar{\theta}_i}$ such that $\theta_1 < \cdots < \theta_N$ and $\bar{\theta}_1 < \cdots < \bar{\theta}_N$. Let $u_{ij} \equiv \bar{\theta}_j - \theta_i$ and $\delta_{ij} \equiv \mathbb{1}_{u_{ij} < 0}$. The squared Cramér distance between the distributions can be expressed as*

$$\int_{-\infty}^{\infty} (F(z) - \bar{F}(z))^2 dz = \frac{1}{N^2} \sum_{i=1}^{N} |u_{ii}| + 2 \left( \sum_{j=i+1}^{N} \delta_{ij}|u_{ij}| + \sum_{j=1}^{i-1}(1 - \delta_{ij})|u_{ij}| \right). \tag{11}$$

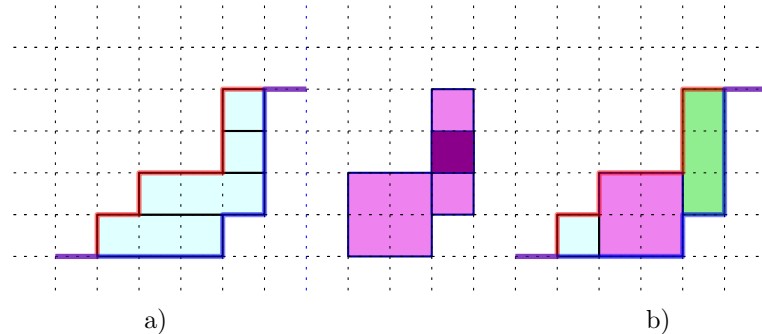

a)                                                                                    b)

Figure 9: **Computing the Cramér distance between $\bar{F}$ (red) and $F$ (blue) with a tiling operator.**
a) starting point represents $\rho_1 = \frac{1}{N^2} \sum_{r \in R_1} u_r$. b) ending point represents the squared Cramér
distance $\frac{1}{N^2} \left( u_1 1^2 + u_2 2^2 + u_3 3^2 \right)$, where $u_i$ is the width of each rectangles in b). Notice that only
the leftmost part of the leftmost rectangle of a) remains in b), the rest has been replaced by taller
rectangles occupying the whole height. The middle diagram illustrates the effect of the tiling operator
$\rho_2$ yielding the final rectangle in the middle and, on the right, two overlapping rectangles—that need
to be replaced by a taller one—and an oversubstracted rectangle in pink. The result of $\rho_1 + \rho_2 + \rho_3$
is shown in b), a rectangle of height 3 has been added, the two overlapping rectangles have been
removed and the pink rectangle has been added back.

*Proof.* In order to compute the squared Cramér distance on a uniform grid, we proceed in a construc-
tive way as follows. The idea is to cover the area between the two curves with rectangular tiles as in
Fig. 9 to compute the integral by pieces. A tile of height $i/N$ and width $u$ corresponds to the term
$u(i/N)^2$. We start from Fig. 9 a) and replace parts of tiles to arrive to b).

Our demonstration unfold through these steps; First, we prove formally that our operator is well built:
the sum of the tiling measured with the operator $\rho$ is equal to the Cramér distance between the two
curves. Secondly, we derive Eq. (11) by using that tiling operator.

First consider an interval $u^+ \equiv [t_1, t_2]$ such that $\bar{F}(t_1) = F(t_1)$, $\bar{F}(t_2) = F(t_2)$ and $\bar{F}(z) >$
$F(z) \; \forall z \in (t_1, t_2)$. Let us define the tiling operator $\rho_h$ for $h \geq 1$

$$\rho_h(F, \bar{F}, u^+) \equiv \sum_{r \in R_h} u_r \left( \frac{h}{N} \right)^2 - 2u_r \left( \frac{h-1}{N} \right)^2 + \mathbb{1}_{h>1} u_r \left( \frac{h-2}{N} \right)^2 \tag{30}$$

$$= \begin{cases} \sum_{r \in R_h} \frac{u_r}{N^2}, & \text{for } h = 1 \\ \sum_{r \in R_h} \frac{2u_r}{N^2}, & \text{otherwise} \end{cases} \tag{31}$$

where $u_r$ is the width of a rectangle $r$ in the set $R_h$ of rectangles of height $h$ whose upper left and
lower right angles are aligned with quantiles of, respectively, $\bar{F}$ and $F$ lying in $u^+$. Note that these
rectangles lie completely within the difference area since $F$ and $\bar{F}$ are monotonically increasing.
Note that $\rho_1$ corresponds to the initial step depicted in Fig. 9 a). Intuitively, for $h > 1$, the operator
replaces parts of width $u_r$ of two tiles of height $h - 1$ by a tile of height $h$ and width $u_r$ and fixes
oversubstracted tiles of the step $h - 2$.

More formally, let us define $\rho^h(F, \bar{F}, u^+) \equiv \sum_{d=1}^{h} \rho_d(F, \bar{F}, u^+)$. We prove by induction the
following property. Given a partition of $u^+$ in a set of intervals $U^+$ such that for any $u \in U^+$,

$$\bar{F}(z) - F(z) = \frac{d_u}{N} > 0 \; \forall z \in u, \tag{32}$$

where $d_u$ depends on $u$ only, then

$$\rho^h(F, \bar{F}, u^+) = \frac{1}{N^2} \sum_{u \in U^+} |u| \left[ \mathbb{1}_{d_u \leq h} d_u^2 + \mathbb{1}_{d_u > h} \left[ (d_u - h + 1)h^2 - (d_u - h)(h-1)^2 \right] \right]. \tag{33}$$

For $h = 1$, in any interval $u$, there are $d_u$ tiles of height 1 that have a non-empty projection on $u$
therefore

$$\rho^h(F, \bar{F}, u^+) = \frac{1}{N^2} \sum_{u \in U^+} |u| d_u \tag{34}$$

426  since $\mathbb{1}_{d_u \leq h} d_u^2 + \mathbb{1}_{d_u > h}(d_u - h + 1)h^2 = d_u$, which validates the base case.

427  For $h > 1$, for each $r \in R_h$, $\rho_h$ adds three terms that can be decomposed in terms that match the
428  segments of $U^+$. By noting that for each interval $u \in U^+$ there will be $\mathbb{1}_{d_u \geq h}(d_u - h + 1)$ rectangles
429  in $R_h$ with non-empty projection on $u$, we have

$$\rho_h(F, \bar{F}, u^+) = \frac{1}{N^2} \sum_{u \in U^+} |u| \mathbb{1}_{d_u \geq h}(d_u - h + 1)\left[h^2 - 2(h-1)^2 + (h-2)^2\right] \tag{35}$$

430  Assuming the property holds for $h - 1$, we have

$$\rho^h(F, \bar{F}, u^+) \tag{36}$$

$$= \rho^{h-1}(F, \bar{F}, u^+) + \rho_h(F, \bar{F}, u^+) \tag{37}$$

$$= \frac{1}{N^2} \sum_{u \in U^+} |u| \left(\mathbb{1}_{d_u \leq h-1} d_u^2 + \mathbb{1}_{d_u > h-1}\left[(d_u - h + 2)(h-1)^2 - (d_u - h + 1)(h-2)^2\right] + \right.$$

$$\left. \mathbb{1}_{d_u \geq h}(d_u - h + 1)\left[h^2 - 2(h-1)^2 + (h-2)^2\right]\right) \tag{38}$$

$$= \frac{1}{N^2} \sum_{u \in U^+} |u| \left(\mathbb{1}_{d_u \leq h-1} d_u^2 + \mathbb{1}_{d_u \geq h}\left[(d_u - h + 1)h^2 - (d_u - h)(h-1)^2\right]\right) \tag{39}$$

$$= \frac{1}{N^2} \sum_{u \in U^+} |u| \left(\mathbb{1}_{d_u \leq h} d_u^2 + \mathbb{1}_{d_u > h}\left[(d_u - h + 1)h^2 - (d_u - h)(h-1)^2\right]\right) \tag{40}$$

431  since $\mathbb{1}_{d_u > h-1} = \mathbb{1}_{d_u \geq h}$ and $\mathbb{1}_{d_u = h}\left[(d_u - h + 1)h^2 - (d_u - h)(h-1)^2\right] = \mathbb{1}_{d_u = h} d_u^2$.

432  Since $\mathbb{1}_{d_u \leq N} = 1 - \mathbb{1}_{d_u > N} = 1$, the final tiling $\rho^N(F, \bar{F}, u^+)$ corresponds to the Cramér distance
433  on the interval $u^+$, i.e.

$$\rho^N(F, \bar{F}, u^+) = \frac{1}{N^2} \sum_{u \in U^+} |u| d_u^2. \tag{41}$$

434  Now, we are going to use (31) to get to the claimed expression. First note that for a rectangle $r \in R_h$
435  with upper leftmost and lower rightmost angles corresponding, respectively, to $\bar{\theta}_j$ and $\theta_i$, its width
436  is $u_r = |u_{ij}|$. Since $\theta_1 < \cdots < \theta_N$ and $\bar{\theta}_1 < \cdots < \bar{\theta}_N$, the condition that $\bar{F}(z) > F(z)$ for
437  such rectangles is equivalent to $\delta_{ij} = 1 \wedge i \leq j$. By symmetry, $\bar{F}(z) < F(z)$ is equivalent to
438  $\delta_{ij} = 0 \wedge j \leq i$. We consider the case $i = j$ separately to avoid double counting and also because it
439  corresponds to $h = 1$. Therefore, from (31), we have

$$\rho^N(F, \bar{F}, \mathbb{R}) = \sum_{r \in R_1} \frac{u_r}{N^2} + \sum_{h=2}^{N} \sum_{r \in R_h} \frac{2u_r}{N^2} \tag{42}$$

$$= \sum_{i=1}^{N} \frac{|u_{ii}|}{N^2} + \sum_{i=1}^{N-1} \sum_{j=i+1}^{N} \delta_{ij} \frac{2|u_{ij}|}{N^2} + \sum_{j=1}^{N-1} \sum_{i=j+1}^{N} (1 - \delta_{ij}) \frac{2|u_{ij}|}{N^2}. \tag{43}$$

440  By taking out common factors and swapping the indices of the two rightmost sums, we get the
441  expression (11). □

442  **Corollary 2.** *For $F(z) \equiv \frac{1}{N} \sum_{i=1}^{N} \mathbb{1}_{z \geq \theta_i}$ and $\bar{F}(z) \equiv \frac{1}{N} \sum_{i=1}^{N} \mathbb{1}_{z \geq \bar{\theta}_i}$ we have*

$$\frac{\partial \mathcal{L}_{\mathrm{QR}}(F, \bar{F})}{\partial \theta_i} = \frac{1}{N}\left(\frac{1 - 2i}{2} + \sum_{j=1}^{N} \delta_{ij}\right) \quad and \quad \frac{\partial \ell_2^2(F, \bar{F})}{\partial \theta_i} = \frac{1}{N^2}\left(1 - 2i + 2\sum_{j=1}^{N} \delta_{ij}\right) \tag{12}$$

443  *where $\delta_{ij} \equiv \mathbb{1}_{u_{ij} < 0}$. Therefore, their gradients are collinear, i.e.*

$$\nabla_{\boldsymbol{\theta}} \mathcal{L}_{\mathrm{QR}} = \frac{N}{2} \nabla_{\boldsymbol{\theta}} \ell_2^2. \tag{13}$$

*Proof.* For a target distribution $\bar{F}(z) = \frac{1}{N}\sum_{i=1}^{N} \mathbb{1}_{z \geq \bar{\theta}_i}$, the quantile regression loss can be expressed as

$$\mathcal{L}_{\mathrm{QR}}(F, \bar{F}) = \sum_{i=1}^{N} \frac{1}{N} \sum_{j=1}^{N} \rho_{\hat{\tau}_i}(\bar{\theta}_j - \theta_i) \tag{44}$$

$$= \frac{1}{N} \sum_{i=1}^{N} \sum_{j=1}^{N} (\bar{\theta}_j - \theta_i)(\hat{\tau}_i - \delta_{ij}) \tag{45}$$

and thus

$$\frac{\partial \mathcal{L}_{\mathrm{QR}}(F, \bar{F})}{\partial \theta_i} = \frac{1}{N} \sum_{j=1}^{N} (\delta_{ij} - \hat{\tau}_i) \tag{46}$$

$$= \frac{1}{N} \left( \frac{1 - 2i}{2} + \sum_{j=1}^{N} \delta_{ij} \right). \tag{47}$$

In order to obtain the partial derivative of the squared Cramér distance, first note that $\delta_{ij}|u_{ij}| = \delta_{ij}(\theta_i - \bar{\theta}_j)$, $(1 - \delta_{ij})|u_{ij}| = (1 - \delta_{ij})(\bar{\theta}_j - \theta_i)$ and $|u_{ii}| = \delta_{ii}(\theta_i - \bar{\theta}_i) + (1 - \delta_{ii})(\bar{\theta}_i - \theta_i)$. By replacing these quantities in (11) and taking the derivative with respect to $\theta_i$ we obtain

$$\frac{\partial \ell_2^2(F, \bar{F})}{\partial \theta_i} = \frac{1}{N^2} \left[ 2\delta_{ii} - 1 + 2 \left( \sum_{j=i+1}^{N} \delta_{ij} + \sum_{j=1}^{i-1} (\delta_{ij} - 1) \right) \right] \tag{48}$$

$$= \frac{1}{N^2} \left( 2 \sum_{j=1}^{N} \delta_{ij} - 1 + 2 \sum_{j=1}^{i-1} (-1) \right) \tag{49}$$

$$= \frac{1}{N^2} \left( 1 - 2i + 2 \sum_{j=1}^{N} \delta_{ij} \right). \tag{50}$$

$\square$

## A.2   Correctness of Algorithm 1

**Proposition 1.** *Given two distributions $F(z) = \frac{1}{N}\sum_{i=1}^{N} \mathbb{1}_{z \geq \theta_i}$, and $\bar{F}(z) = \frac{1}{N}\sum_{i=1}^{N} \mathbb{1}_{z \geq \bar{\theta}_i}$, Algorithm 1 computes*

$$\int_{-\infty}^{\infty} (F(z) - \bar{F}(z))^2 dz = \sum_{i=1}^{2N-1} (\theta'_{i+1} - \theta'_i) \left( \sum_{j \text{ s.t. } \theta_j \leq \theta'_i} \frac{1}{N} - \sum_{j \text{ s.t. } \bar{\theta}_j \leq \theta'_i} \frac{1}{N} \right)^2. \tag{51}$$

*Proof.* Consider the sorted sequence of merged quantiles

$$\boldsymbol{\theta'} \equiv \theta'_1, \ldots, \theta'_{2N} \equiv \mathrm{sort}\left( \{\theta_i\}_{i=1..N} \bigcup \{\bar{\theta}_i\}_{i=1..N} \right). \tag{52}$$

We have that $F(z) - \bar{F}(z) \equiv \Delta_i$ is constant in $[\theta'_i, \theta'_{i+1}), \forall i \in 1..2N$. Therefore,

$$\int_{-\infty}^{\infty} (F(z) - \bar{F}(z))^2 dz = \sum_{i=1}^{2N-1} \int_{\theta'_i}^{\theta'_{i+1}} (F(z) - \bar{F}(z))^2 dz = \sum_{i=1}^{2N-1} (\theta'_{i+1} - \theta'_i) \Delta_i \tag{53}$$

If $\theta'_i \leq z < \theta'_{i+1}$, then

$$F(z) = \frac{1}{N} \sum_{j=1}^{N} \mathbb{1}_{z \geq \theta_j} = \frac{1}{N} \sum_{j \text{ s.t. } \theta_j \leq \theta'_i} 1 \tag{54}$$

$$\bar{F}(z) = \frac{1}{N} \sum_{j=1}^{N} \mathbb{1}_{z \geq \bar{\theta}_j} = \frac{1}{N} \sum_{j \text{ s.t. } \bar{\theta}_j \leq \theta'_i} 1 \tag{55}$$

and thus

$$\Delta_i = \sum_{j \text{ s.t. } \theta_j \leq \theta'_i} \frac{1}{N} - \sum_{j \text{ s.t. } \bar{\theta}_j \leq \theta'_i} \frac{1}{N}, \tag{56}$$

which proves (51).

The algorithm computes the differences $(\theta'_{i+1} - \theta'_i)$ and stores them in $\Delta_z$. After the steps

$$\Delta_\tau \leftarrow \text{concat}\left(-\frac{1}{N}\mathbf{1}_N, \frac{1}{N}\mathbf{1}_N\right) \tag{57}$$

$$\Delta_\tau \leftarrow \Delta_\tau[i_1, \ldots, i_N], \tag{58}$$

in words, the $i$-th element of the vector $\Delta_\tau$ is $-1$ if $\theta'_i$ comes from $\bar{\theta}$ and 1 otherwise, i.e.

$$\Delta_\tau[i] = (-1)^{\mathbb{1}_{\exists j \theta'_i \equiv \bar{\theta}_j}} \tag{59}$$

where $\equiv$ denotes symbol equality. After the final step

$$\Delta_\tau \leftarrow \text{cumsum}\,(\Delta_\tau)\,[:\text{-1}], \tag{60}$$

the $i$-th element of the vector $\Delta_\tau$ can be expressed as

$$\Delta_\tau[i] = \frac{1}{N}\sum_{k=1}^{i}(-1)^{\mathbb{1}_{\exists j \theta'_k \equiv \bar{\theta}_j}}. \tag{61}$$

If $\theta'_i \neq \theta'_{i+1}$, then $\Delta_\tau[i] = \Delta_i$. Otherwise, $\Delta_\tau[i] \neq \Delta_i$, but, since $\theta'_{i+1} - \theta'_i = 0$, the corresponding term in (51) is zero too. Therefore, the algorithm produces the claimed output. $\square$

### A.3 Hyperparameters

All the experimental results for CNC-CR-DQN, NC-CR-DQN and CR-DQN were obtained with the hyperparameters' values shown in Table 1, which are the same that those used to generate the QR-DQN results provided by DQN_ZOO. The last two hyperparameters are specific to CNC-QR-DQN and NC-CR-DQN.

### A.4 Additional experimental results

Fig. 10 shows the online training performance of CNC-CR-DQN in comparison to the pure distributional contenders C51, QR-DQN and IQN, on the full Atari-57 benchmark.

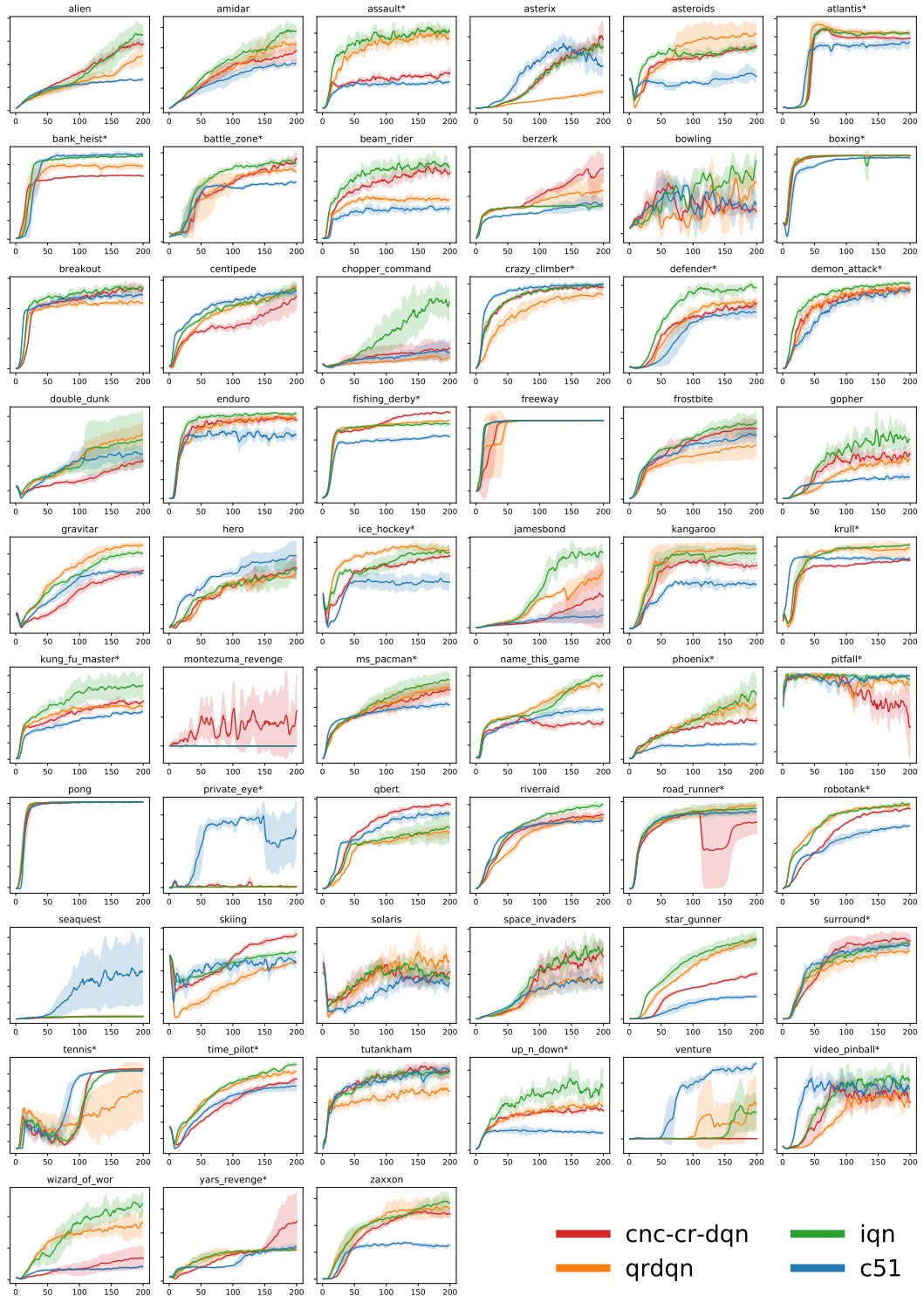

Figure 10: **Training performance on the Atari-57 benchmark.** Curves are averages over a number of seeds, smoothed over a sliding window of 5 iterations, and error bands give standard deviations. For C51, QR-DQN and IQN, 5 seeds were used (provided by DQN_ZOO [24]). For CNC-CR-DQN, 3 seeds were used for all the games except those indicated by ∗, for which only 2 seeds were used.

Table 1: **Hyperparameters used in our** `DQN_ZOO` **implementation of CNC-CR-DQN.**

| Hyperparameter | Value | Comment |
|---|---|---|
| replay_capacity | 1e6 | |
| min_replay_capacity_fraction | 0.05 | Min replay set size for learning |
| batch_size | 32 | |
| max_frames_per_episode | 108000 | = 30 min |
| num_action_repeats | 4 | In frames |
| num_stacked_frames | 4 | |
| exploration_epsilon_begin_value | 1 | |
| exploration_epsilon_end_value | 0.01 | |
| exploration_epsilon_decay_frame_fraction | 0.02 | |
| eval_exploration_epsilon | 0.001 | |
| target_network_update_period | 4e4 | |
| learning_rate | 5e-5 | |
| optimizer_epsilon | 0.01 / 32 | ADAM's parameter |
| additional_discount | 0.99 | Discount_rate multiplier |
| max_abs_reward | 1 | |
| max_global_grad_norm | 10 | |
| num_iterations | 200 | |
| num_train_frames | 1e6 | Per iteration |
| num_eval_frames | 5e5 | Per iteration |
| learn_period | 16 | One learning step each 16 frames |
| num_quantiles | 201 | $N$ |
| Convolutional layer 1 | 32, (8, 8), (4, 4) | num_features, kernel_shape, stride |
| Convolutional layer 2 | 64, (4, 4), (2, 2) | |
| Convolutional layer 3 | 64, (3, 3), (1, 1) | |
| n_layers | 1 | Number of hidden layers $\lambda$ |
| n_nodes | 512 | Number of nodes $\eta$ per hidden layer |