# OpenReview forum: "A Cramér Distance perspective on Non-crossing Quantile Regression in Distributional Reinforcement Learning"
_NeurIPS.cc/2021/Conference — NeurIPS 2021 Submitted_

### Official Review · Reviewer_rysK · 2021-07-04

**Rating:** 5
**Confidence:** 4

**Summary:**

This paper proposed to apply the Cramér distance to replace the 1-Wasserstein distance in the class distributional reinforcement learning. To justify the validity of this update, this paper proved two key properties: 1) Cramér projected distributional Bellman operator is a contraction and, 2) QR loss and Cramér gradients are collinear under non-crossing constraints. This paper also designed a low-complexity algorithm for computing the Cramér distance and a centered non-crossing architecture to implement the proposed algorithms.

**Ethical Concerns:**

No.

**Limitations And Societal Impact:**

Societal impact: The authors believe their work has no negative societal impact, I generally agree that their negative impact is limited, but I think the goal of RL is to develop autonomous agents that can replace humans in many positions, for example, drivers. The author might want to discuss the influence of autonomous agents on the job market.

Limitations: The discussion of limitations should be expanded, especially about why a clear win is not observed in the Atari benchmark.

**Main Review:**

The idea of this paper is novel and not incremental. It proposed a new framework for modeling the distance between distribution in the distributional Bellman operator. Given that most of the recent algorithms for distributional RL are based on QR, this paper is a significant plus to this field. The major issue of this paper is that the author(s) generally can not convince me that the Cramér-based models can outperform the QR-based models. The proofs generally guarantee that CR-DQN can learn reasonable distributions of Q values because it resembles QR-DQN, but none of them can help readers understand where is the improvement. This paper lists some benefits like 1) CR enables removing a constraint (line 172-174) and 2) extreme quantiles are hard to estimate in the classic QR-DQN (line 186), but none of them is fully defined or evaluated. The motivation is vague from my point of view. The empirical results do not show a clear win of CNC-CR_QDN and the results are not explained.

Detail comments:

1) Line 185 "As shown by the classical theory of extreme values", could you explain more to help readers understand it? since it is a major motivation for your network architecture. By the way, figure 4 in [31](NC-QR_DQN) has claimed NC-QR_DQN converged faster than QR_DQN, which make me believe the reduction of training efficiency (from the "symmetries" (line 177)) in CR is significant and adding the CNC structure can not resolve this issue. It explains why CR-DQN is generally more sample inefficient than the QR-based methods. Your results in Figure 7 and the appendix generally confirm this problem.

2) The Atari 57 experiment should include both NC-QR-DQN and NC-CR-DQN as baselines since the author(s) proposed replacing the original non-crossing network with the CNC architecture. Comparing both NC-QR-DQN and NC-CR-DQN can provide a form of ablation study that helps readers understand the influence of each component (CR and CNC).



**Time Spent Reviewing:**

6

---

> ### Author Response · Authors · 2021-08-09
> **Response to Reviewer rysK**
>
> Thank you for your comments and suggestions.
>
> *"The major issue of this paper is that the author(s) generally can not convince me that the Cramér-based models can outperform the QR-based models."* : we are not claiming that. In particular, our theoretical results show that any non-crossing network should give the same results with QR and Cramér losses, since the gradients are collinear (in particular, we could informally write CNC-CR-DQN = CNC-QR0-DQN  and NC-CR-DQN = NC-QR0-DQN where QR0=QR loss without Huberization).
>
> *Detailed comments:*
> [31] claimed that NC-QR-DQN converged faster than QR-DQN **in the exploration bonus setting**. In that paper, no results are provided for the standard $\varepsilon$-greedy setting that we consider. Since the cost of the experiments is high, we couldn't afford running their network with the standard $\varepsilon$-greedy strategy on the whole set of games. The examples of Figure 5, where NC-CR-DQN underperforms significantly, motivated us to propose a new non-crossing architecture to test our theoretical findings which, from our point of view, are the most important contribution of this paper.

---

> > ### Comment · Reviewer_rysK · 2021-09-01
> > **Acknowledging the author response**
> >
> > Thanks for responding to my review. Your general responses resolve some of my concerns and they should be present more clearly in the main paper. However, I  still believe the motivation of studying Cramér-based models is limited in the sense that the benefit of Cramér distance is not fully defined and studied. It is unclear what's the gain of replacing Wasserstein with Cramér in terms of building a more advanced distributional RL model. I will main my initial score.

---

### Official Review · Reviewer_mDzE · 2021-07-12

**Rating:** 4
**Confidence:** 4

**Summary:**

This paper draws an equivalence between 1-Wasserstein distance and Cramer distance on staircase distributions, use Cramer distance in the place of 1-Wasserstein (and quantile regression) to avoid order constraints among the particles $\theta_i$ and consequently, proposes a non-crossing neural net for computing the Cramer distance in DRL.

**Ethical Concerns:**

No concern.

**Limitations And Societal Impact:**

The authors have not adequately addressed the limitations of their work.

**Main Review:**

**Strong points**:
-	The connection between 1-Wasserstein and Cramer seems novel and interesting in DRL.  Especially, quantile regression in QR-DQN imposes order into the learnable particles $\\{ \theta_i \\}_{i=1}^N$ as each $\theta_i$ is an estimate of the $\hat{\tau}_i$-quantile which implies the monotonicity in the particles, i.e., $\theta_1 \leq \theta_2 \leq ... \leq \theta_N$. Cramer distance on the other hand is equivalent to 1-Wasserstein in terms of projection onto the space of a mixture of Heaviside step functions and in terms of gradients. Moreover, Cramer distance does not require order among $\theta_i $

**Weak points**:
- Many parts of the writing are unclear and need more improvement (see below)
- I think the present paper lacks a discussion and comparison with related work that is directly related to the problem and proposed solution.
- The motivation and purpose of the experimental section are unclear and there is no conclusion or discussion drawn from the experiment. The experimental result does not show that the proposed algorithm CNC-CR-DQN outperforms QR-DQN with statistical significance in the whole range of Atari games (Fig 6). This raises the question of whether the proposed algorithm is effective and necessary.
- The theoretical results are quite simple and the connection between Cramer and Wasserstein is already made in [2].


**Comments and Questions**:
-	I think the paper lacks relevant literature. Most distributional RL methods are based on quantile regression that requires ordered statistics in neural representation, creating challenges such as maintaining ordered statistics, monotonicity, and “crossing quantiles”. I think a recent work [Nguyen-Tang et al AAAI’21, “Distributional Reinforcement Learning via Moment Matching”] can completely avoid such problems in quantile-regression-based methods using moment matching; thus it is directly related to the problem and the proposed approach in the present paper. What is the difference between your approach and Nguyen-Tang's in addressing the order statistics constraint in QR-based DRL?
-	The paper addresses the crossing quantile problem in DRL but it only briefly touches on crossing quantile in one sentence at line 47-48 and at line 155. I think it is inadequate. More discussion is needed and probably one dedicated paragraph for this is necessary.
-	Line 130-131: What exactly permutation in the symmetric group of size N? What **symmetric group** are you referring to? This part needs to be clearer.
-	Eq. (14): What is $\theta’$? A typo?
-	The experiment section lacks a clear explanation what is the purpose of the experiment? What conclusions can be drawn from the results? Currently, this section merely just describes the experimental procedure without insights drawn from it.
-	I think putting markers on Fig 6 would make it easier to read. Some colors are very similar.
-	Fig 6:  CNC-CR-DQN outperforms QR-DQN in several games but in the entire Atari set, they perform very similarly (Fig. 6). How can you conclude that the proposed method that draws on Cramer distance without crossing quantiles works better in a statistical sense?
- 1-Cramer distance ($ \int |F(z) - G(z)| dz $) is identical to 1-Wasserstein distance and using 1-Crammer distance also does not require order among $\theta_i$ as 2-Cramder distance. Why don't just use 1-Cramer distance?
- What kind of games from the Atari set the proposed algorithm performs better than QR-DQN?

**Initial Recommendation**
While the idea of using Cramer distance as an alternative to the quantile regression loss is interesting, I do not find sufficient novelty and/or significant empirical performance in the current form, raising the question about the necessity and effectiveness of the proposed algorithm; moreover, the writing seems in rush and lacks related work. I vote for reject this time for the current form.


**Time Spent Reviewing:**

5

---

> ### Author Response · Authors · 2021-08-09
> **Response to Reviewer mDzE**
>
> Thank you for your comments and suggestions.
>
> *Weak points:*
> * *"I think the present paper lacks a discussion and comparison"*: please see general response
> * *"The motivation and purpose of the experimental section are unclear"*: please see general response
> * *"The theoretical results are quite simple and the connection between Cramer and Wasserstein is already made in [2]."*: our results are not trivial (see proofs in Appendix) and, as far as we know, are established in this work for the first time. [2] considers both distances and shows the the unbiasedness of Cramér gradients but don't establish the connections proved in our work.
>
> *Detailed comments and questions:*
> 1. *"Most distributional RL methods are based on quantile regression that requires ordered statistics"* and *"The paper addresses the crossing quantile problem in DRL"*: our theoretical results elucidate connections between Cramér and QR losses under the assumption of non-crossing quantiles. However, as explained in lines 171-174, our Cramér algorithm does not require ordered statistics (thanks to the sort operation). However, as shown in Fig 5 and discussed in line 177, without this order constraint the learning process can be more difficult. We left for future work to investigate how we can improve the learning process with the Cramér loss without order constraints. Thanks for pointing out this interesting recent work, we will add a reference to it.
> 2. Line 130-131: we will add as a footnote "the finite symmetric group $S_n$  defined over a finite set of  symbols consists of the all permutations that can be performed on the $n$ symbols."
> 3. Eq. (14): $\theta'_i$ is the $i$-th element of the vector θ' (defined in line 158), this will be clarified in the final version.
> 4. *"The experiment section lacks a clear explanation..."*: please see general response
> 5. *"CNC-CR-DQN outperforms QR-DQN in several games but in the entire Atari set..."*: please see general response
> 6. *"...Why don't just use 1-Cramer distance?"* : as said in lines 8 and 44 (shown in [2])  1-Wasserstein distance has biased gradients. This is what motivates the use of the QR loss and the $\ell_2$ (Cramér) distance.

---

> > ### Comment · Reviewer_mDzE · 2021-08-30
> > **Acknowledging the author response**
> >
> > Thank you the authors for the response to my concern. However, my concerns about the experimental significance and technical novelty of this paper remain as they seem more inherent limitations of the proposed method. Thus, I will maintain my initial score. I encourage the authors to take the discussion here into account to strengthen this direction further.

---

### Official Review · Reviewer_fTr8 · 2021-07-13

**Rating:** 4
**Confidence:** 4

**Summary:**

The authors provide theoretical analysis of distributional RL under Cramer distance. The authors further propose a neural architecture that yields monotonic quantiles and achieved improvements over prior works.

**Limitations And Societal Impact:**

The authors discuss the limitations adquately in their work.

**Main Review:**

Originality: Low. Cramer distance in dsitributional RL has been studied in prior works, the authors provide some missing theoretical results. The proposed algorithm is slightly modified from NC-QR-DQN.

Quality: Medium. The theories are correct, but the experiment results are insignificant.

Clarity: Medium. The paper is easy to read, except for some minor issues.
 1. l.154 Formula...
 2. In l.92, l_p is the p-Wasserstein distance, but later used as the Cramer distance without clarification.
 3. Figure 3. is a little hard to interpret

Significance: Low.

Major concerns:
 1. After reading the paper, I find that while correct, the theoretical results are not likely to be able to improve dsitributional RL algorithms.
 2. From my opinion, NC-QR-DQN should be an important baseline of the work. But the authors didn't provide further comparison other than the three cases in Figure.5, therefore it is hard to tell the actual improvements of the proposed neural architecture.
 3. Missing score table, incomplete experiments (2 seeds) in Figure 10.


Questions & suggestions:
 1. If Cramer loss and Quantile Regression loss yields collinear gradients, why not use QR loss to save the trouble of computing Cramer distance? We only need the gradients to update network parameters.
 2. According to my understanding of the paper, the proposed neural architecture, CNC, have three major differences compared with NC-QR-DQN:
  1). For subnetwork 1, outputs median quantile value instead of minimum quantile value. Then, substract the cumsum of the first half per-quantile value differences instead of add.
  2). Remove the scaling factor, thereby using ReLU instead of Softmax for per-quantile value difference.
    I would suggest an ablation study on those two modifications to further investigate which one contributes more to the improvements.

**Time Spent Reviewing:**

5 hours

---

> ### Author Response · Authors · 2021-08-09
> **Response to Reviewer fTr8**
>
> Thank you for your comments and suggestions.
>
> *major concerns:*
> * *"1. After reading the paper, I find that while correct, the theoretical results are not likely to be able to improve dsitributional RL algorithms."*:
> We respectfully disagree with this opinion. As we say in our general response, we believe that future research can build on these new results to make new DRL theory and algorithms related to the Cramér distance, the 1-Wassertein distance or the QR loss. Time will tell…
>
> * *"2. From my opinion, NC-QR-DQN should be an important baseline of the work."* : please see our general response
>
> * *"3. Missing score table, incomplete experiments (2 seeds) in Figure 10:"* this will be fixed in the final version of the paper, please see our general response.
>
> *minor issues:*
> * Line 92 says that the $p$-Wasserstein distance  is the $\ell_p$ metric between the **inverse** CDFs. The Cramér distance is the $\ell_2$ metric between the CDFs. We don't see any notation issue here.
>
>
> *questions and suggestions:*
> 1. Yes, the QR and the Cramér losses give collinear gradients so they are exchangeable for gradient descent algorithms, in the fixed quantile levels setting. We used the Cramér loss to illustrate this fact since QR is the commonly used loss. In fact, it is no trouble to compute the Cramér loss since our algorithm has $O(N \log N)$ complexity while for the QR loss  algorithm it is $O(N^2)$.
> 2. Thanks for the suggestion. As we said in the general response, the goal of this paper was to present the theoretical results. The experimental results aim at illustrating the theory. Since we found that NC-QR-DQN drastically fails in some cases, we proposed this new architecture to enforce the non-crossing constraints. We agree that more research is required to understand the effect of the different components of the architecture, but we leave this for another paper.

---

### Official Review · Reviewer_P97K · 2021-07-16

**Rating:** 3
**Confidence:** 5

**Summary:**

This work proposes a Cramer loss for the training process of distributional RL, and design a centered non-crossing architecture for distributional RL.

**Limitations And Societal Impact:**

Yes

**Main Review:**

Strong points:
1.	Clarity: The paper is well structured.
2.	Technical novelty: A efficient algorithm is proposed to compute the Cramer distance.

Weak points:
1.	I am concerned about the validity of the experiment results, since the performance of QR-DQN in Figure 5 is much worse than the results in their original paper. I am not sure if the authors correctly implement this method and the conclusion seems to be unconvincing.
2.	The advantages of Cramer loss over quantile (Huber) loss are not well presented. As mentioned in your paper, both losses achieve unbiased gradient and equivalence to 1-Wasserstein in minimization problem. What are the pros of Cramer loss? Computation efficiency? Or else? Corresponding ablation studies are lacked.
3.	Unfortunately, there is a nearly same network architecture called NDQFN (Zhou et al., 2021) with proposed centered non-crossing architecture (CNC), except that CNC focus on median while NDQFN aims at \tau = 0. However, NDQFN seems to achieve much better performance than CNC with Cramer loss.
4.	The experimental results in the present form is not strong. From the Figure 6 and Figure 10 in Appendix, it seems CNC-CR-DQN perform comparable with QR-DQN, even slightly worse.

Hence, I vote for rejection for current form.

Mentioned reference:
Zhou, F.,Zhu, Z., Kuang, Q., Zhang, L*. Non-decreasing quantile function network withefficient exploration for distributional reinforcementlearning. International Joint Conference onArtificial Intelligence (IJCAI), 2021. In press.


**Time Spent Reviewing:**

3

---

> ### Author Response · Authors · 2021-08-09
> **Response to Reviewer P97K**
>
> Thank you for your comments and suggestions.
>
> * *"1. I am concerned about the validity of the experiment results"*: Regarding the implementation of QR-DQN: the results come from the precomputed results of Deepmind's DQN_ZOO implementation (https://github.com/deepmind/dqn_zoo/blob/master/results.tar.gz). If you look at  Figure 4 in [10] right plot (training performance), the median (over 3 seeds) is 127% (measured from the plot with https://apps.automeris.io/wpd/) which is close to (and **below**) the 129% found with DQN_ZOO (with 5 seeds).
>
> * *"2 .The advantages of Cramer loss over quantile (Huber) loss are not well presented.."*: Yes, the complexity of our Cramér **algorithm** is better than the QR loss one ($O(N \log N)$ vs $O(N^2)$ ). An important message of our paper is that the standard QR loss is essentially equivalent to the Cramér loss, under non-crossing constraints.
>
> * *"3. Unfortunately, there is a nearly same network architecture called NDQFN… However, NDQFN seems to achieve much better performance than CNC with Cramer loss."*:   Thanks for pointing this recent work out, we will add a reference to it. However, we can't directly conclude from that paper that it achieves a better performance than CNC-CR-DQN,  since, for the whole set of 57 games, only evaluation metrics under the best agent protocol are presented while our results are expressed in terms of online training performance as advised by [20].
>
> * *"4. The experimental results in the present form is not strong."*: please see general response

---

### Author Response · Authors · 2021-08-09
**General response**

First of all, we want to thank the reviewers for their work and efforts.

We would like to clarify the message and the value of our paper from our point of view.
Our work presents novel theoretical results, in the setting of fixed quantile levels, showing that the Cramér loss is equivalent (in terms of minimizer and gradients) to the QR-loss  if we impose a non-crossing constraint. We believe that future research can build on our results to make new DRL theory and algorithms related to the Cramér distance, the 1-Wassertein distance or the QR loss.

The experimental part illustrates the theoretical results, showing that is possible to use the Cramér loss in practice, with non-crossing constraints imposed by the network. In order to clarify this, we will add the following sentence at the beginning of Section 6:
"The goal of this section is to validate our theoretical results and show the practical suitability of the Cramér loss implemented by our algorithm to train an RL agent defined by our CNC-CR-DQN architecture."

 We considered using the already existing NC-QR-DQN network for this purpose but we found that it can totally fail with the standard $\varepsilon$-greedy exploration strategy (see Fig 5) (note that the original paper [31] shows results for NC-QR-DQN with a non-standard exploration strategy only). This motivated our architecture CNC-CR-DQN. We didn't run NC-QR-DQN with the standard $\varepsilon$-greedy strategy over the full set of games for computational budget reasons.

We agree that our experimental results don't allow to conclude that CNC-CR-DQN is in *general* superior to QR-DQN&mdash;and this is not claimed in our paper.
It clearly depends on the game and the number of training iterations performed.
After the submission, we finished running the 57 games with three seeds and the new Fig 6 shows CNC-CR-DQN  even higher in the final iterations.
At iteration 200, the median human-normalized training score for CNC-CR-DQN over the 57 games is 138% while for QR-DQN it is 129%. Fig 6 and Fig 10 will be updated in the final version of the paper showing the results with three seeds for all the games.
Even if we can't say that CNC-QR-DQN is superior to QR-DQN in general, we believe that it is important to show our results for transparency and to help future research on non-crossing architectures, whose performance is quite sensitive to the design choices as shown in Fig 5. The whole output of our runs will be provided in a github repository for easy comparison.

---

### Decision · Program_Chairs · 2021-09-27

**Decision:**

Reject

**Comment:**

As the authors raised concerns with the reviews, I have thoroughly read the paper and even discussed the situation with the senior area chair. While the reviews are all against acceptance, my individual assessment is much more positive. Not to burry the lead, but, after discussions I will be suggesting that the paper not be accepted.

I very strongly encourage the authors to resubmit. Rather than viewing this work as a competitor with NC-QR-DQN, I see this paper as providing some very interesting theoretical connections between all QR-based methods and the Cramer distance. The reviewers are, perhaps rightly, less than confident in the impact of this work due to the empirical results in the paper. This is likely amplified by the discrepancy between the published NC-QR-DQN results (with exploration bonus) and those in this paper (with epsilon-greedy, and also using the Cramer loss?).

However, the other point reviewers have raised, which I agree is less easily dismissed, is that if these losses are essentially equivalent, then how can this have real impact beyond what we already can be done with pure QR? However, most QR-DQN-like methods do actually use the Huber quantiles, while this method is equivalent to a true (non-crossing) QR. Perhaps there is a unique advantage to the non-crossing quantiles in the pure QR case. This seems like something that could be checked by keeping as much the same as in this paper but using the QR loss (without Huber). I think that comparing with the actual NC-QR-DQN algorithm (with the Huber quantiles and architecture, but without the exploration bonus) would be an important addition. I would also suggest improving the clarity of Figures 1 & 2 (right panel), but view this as a fairly minor issue.

Although perhaps worth discussing, I do not think that NDQFN is a necessarily empirical comparison. Obviously reviewers may disagree, and thus to buffer your chances of acceptance it may be worth attempting nonetheless. I would have very much liked to accept this paper, but at the same time I do not think it is quite strong enough to justify completely disregarding all four reviewers feedback. However, please know that your concerns with the reviews have been noted.